# Effects of tetracycline on myocardial infarct size in obese rats with chemically-induced colitis

Yury Yu Borshchev[1,2], Sarkis M. Minasian[1,3], Inessa Yu Burovenko[2,4], Victor Yu Borshchev[5], Egor S. Protsak[1,3], Natalia Yu Semenova[1], Olga V. Borshcheva[1,2], Michael M. Galagudza[1,3]*

1 Institute of Experimental Medicine, Almazov National Medical Research Centre, Saint Petersburg, Russian Federation, 2 Scientific Research Center "Probiocode SP", Moscow, Russian Federation, 3 Department of Pathophysiology, Saint Petersburg Pavlov State Medical University, Saint Petersburg, Russian Federation, 4 Department of Physiology and Sanocreatology, Shevchenko Transnistria State University, Tiraspol, Republic of Moldova, 5 Department of Microelectronics and Biomedical Engineering, Technical University of Moldova, Chisinau, Republic of Moldova

* galagudza@almazovcentre.ru

**Data Availability Statement:** All relevant data are within the manuscript and its Supporting Information files.

## Abstract

### Background

Recent evidence suggests that antibiotic-induced changes in the composition of intestinal microflora, as well as the systemic immunoendocrine effects that result from them, can modulate myocardial tolerance to ischemia-reperfusion injury. The aim of this study was to investigate the effects of tetracycline (TTC) on myocardial infarct size in the isolated hearts obtained from obese rats with chemically-induced colitis (CIC). The association between TTC-induced changes in infarct size and intestinal microbiome composition as well as plasma levels of cytokines and short-chain fatty acids (SCFAs) was also studied.

### Methods

Obesity was induced in Wistar rats by feeding them a high-fat, high-carbohydrate diet for five weeks. A single rectal administration of 3% acetic acid (2 mL) to the rats resulted in CIC. Healthy rats as well as obese rats with CIC received TTC (15 mg daily for 3 days) via gavage. The rats were euthanized, after which isolated heart perfusion with simulated global ischemia and reperfusion was performed. Infarct size was determined histochemically. Lipo-polysaccharide (LPS) and cytokine levels in plasma were measured by enzyme-linked immunosorbent assay, whereas SCFA levels in plasma were measured by gas chromatography/mass spectrometry. The intestinal microbiome was analyzed using reverse transcription polymerase chain reaction.

### Results

The treatment with TTC resulted in significant infarct size limitation (50 ± 7 vs. 62 ± 4% for the control mice, $p < 0.05$) in the hearts from intact animals. However, infarct size was not different between the control rats and the obese rats with CIC. Furthermore, infarct size was

**Funding:** This study was supported by the Russian Science Foundation (project 18-15-00153). The funder had no role in study design, data collection and analysis, decision to publish, or preparation of the manuscript.

**Competing interests:** The authors have declared that no competing interests exist.

significantly larger in TTC-treated obese rats with CIC than it was in the control animals (77 ± 5%, $p < 0.05$). The concentrations of proinflammatory cytokines and LPS in serum were elevated in the obese rats with CIC. Compared to the control rats, the rats with both obesity and CIC had lower counts of *Lactobacillus* and *Bifidobacterium spp.* but higher counts of *Escherichia coli*. The effects of TTC on infarct size were not associated with specific changes in SCFA levels.

## Conclusions

TTC reduced infarct size in the healthy rats. However, this effect was reversed in the obese animals with CIC. Additionally, it was associated with specific changes in gut microbiota and significantly elevated levels of cytokines and LPS.

## Introduction

Antimicrobial drugs are extensively used to treat and prevent bacterial infections. However, they can enter the human body when contaminated food and water are consumed due to the increasing use of antimicrobials in the agricultural industry, as well as when drug manufacturing industries pollute the environment with antibiotics [1,2]. Some antimicrobial agents can have negative effects on the heart. For example, macrolides and fluoroquinolones can increase the incidence of arrhythmias in high-risk patients because they cause prolongation of the QT interval [3,4]. Furthermore, the use of anthracycline antibiotics for treating cancer is associated with significant cardiotoxicity. The results of a previous study showed that 48% of patients who were treated with doxorubicin at a dose of 700 mg/m$^2$ experienced massive cardiomyocyte death and developed chronic heart failure [5]. It is also reported that fungicidal drugs belonging to the echinocandin family can cause cardiotoxicity [6].

Over the past decade, there has been increasing evidence that some antibiotics have beneficial effects on the heart. Some antimicrobials can reduce myocardial ischemia-reperfusion injury (IRI) by having a direct effect on cardiomyocytes or by modifying intestinal microflora, which results in systemic metabolic and neuroimmunoendocrine changes. In a previous study in rats, vancomycin or the probiotic strain *Lactobacillus plantarum 299V* significantly lowered plasma leptin level and reduced infarct size by 27 and 29%, respectively [7]. In a further study, it was demonstrated that increased myocardial tolerance to IRI induced by vancomycin or a mixture of streptomycin, neomycin, bacitracin, and polymyxin B is associated with specific changes in intestinal microbiota [8]. The exact mechanism underlying antibiotic-induced cardioprotection can be investigated by studying the anti-ischemic effects of tetracycline (TTC) in canine and murine models of IRI [9]. TTC antibiotic minocycline treatment demonstrated anti-ischemic effect in animal models of myocardial [10], hepatic [11], and cerebral IRI [12] due to its anti-inflammatory, anti-apoptotic, and antioxidant activity. Furthermore, oral administration of the macrolide antibiotic azithromycin results in reduced myocardial inflammation and improved left ventricular function after infarction in mice, possibly due to an increased number of M2 macrophages [13]. These findings provide a rationale for a new concept of myocardial IRI control and may have applications in the treatment of ischemic heart disease. However, all the above-mentioned studies were performed in young healthy animals. Many cardioprotective therapies have failed in translation because their potencies are significantly reduced or even abolished in the presence of comorbidities [14]. Clinically, ischemic heart disease is most commonly associated with abdominal obesity and both acute and chronic

inflammatory disorders. With this in mind, we investigated the effects of TTC on myocardial IRI in the isolated hearts of rats with diet-induced obesity (DIO) and chemically-induced colitis (CIC). Possible associations between the effects of TTC on cardiac IRI and intestinal microbiome profiles as well as the plasma levels of proinflammatory cytokines, lipopolysaccharide (LPS), and short-chain fatty acids (SCFAs) were also studied.

Our findings have confirmed the infarct-limiting effect of TTC in healthy rats. However, the coexistence of DIO and CIC reversed the cardioprotective effect of TTC. Infarct size was significantly larger in the hearts obtained from TTC-treated obese animals with CIC than it was in the hearts of control rats. This effect of TTC was not associated with changes in SCFA levels but may be attributed to a specific profile of gut microbiota, the extent of endotoxemia, or cytokine imbalance. Our findings support the cautious use antimicrobial agents in patients with comorbidities who also have a high risk of acute coronary syndrome.

## Materials and methods

### Ethics statement

All procedures were performed in accordance with the Guide for the Care and Use of Laboratory Animals (NIH publication No. 85–23, revised 1996) and the European Convention for the Protection of Vertebrate Animals used for Experimental and other Scientific Purposes. The Institutional Animal Care and Use Committee at the Almazov National Medical Research Centre approved the study protocol (Protocol Number 17-DI-1; May 30, 2017). All efforts were made to protect the animals and minimize their suffering during the study. The experiments complied with the ARRIVE guidelines (http://www.nc3rs.org/ARRIVE).

### Animals

Male Wistar rats (age, 11–12 weeks; weight, 300–320 g) were obtained for the study. The animals were maintained under a 12/12-h light/dark cycle and given free access to food and water.

### Reagents

All chemicals used in the experiments were of analytical grade and were purchased from Sigma-Aldrich (St. Louis, MO, USA) unless otherwise specified.

### Development of DIO in the rats

DIO was developed in the animals by feeding them a high-fat, high-carbohydrate diet (HFCD; 40% fat, 45% sucrose; Dyets Inc., Bethlehem, PA, USA) for five weeks prior to heart isolation [15].

### Induction of colitis

Colitis was induced in rats with DIO via a single rectal administration of 3% acetic acid (2 mL) using a polyethylene tube (2 mm in diameter), which was inserted into the rectum to a depth of 5.5 cm [16]. The rats were positioned for the hind limbs to be elevated above the head during the rectal instillation and for 1 min afterwards to prevent leakage of the solution.

### Experimental design

The animals were randomly put into the following five groups (Fig 1): control (CON, n = 14, each rat received 1 mL of the vehicle), TTC (n = 13, each rat received 15 mg TTC in 1 mL

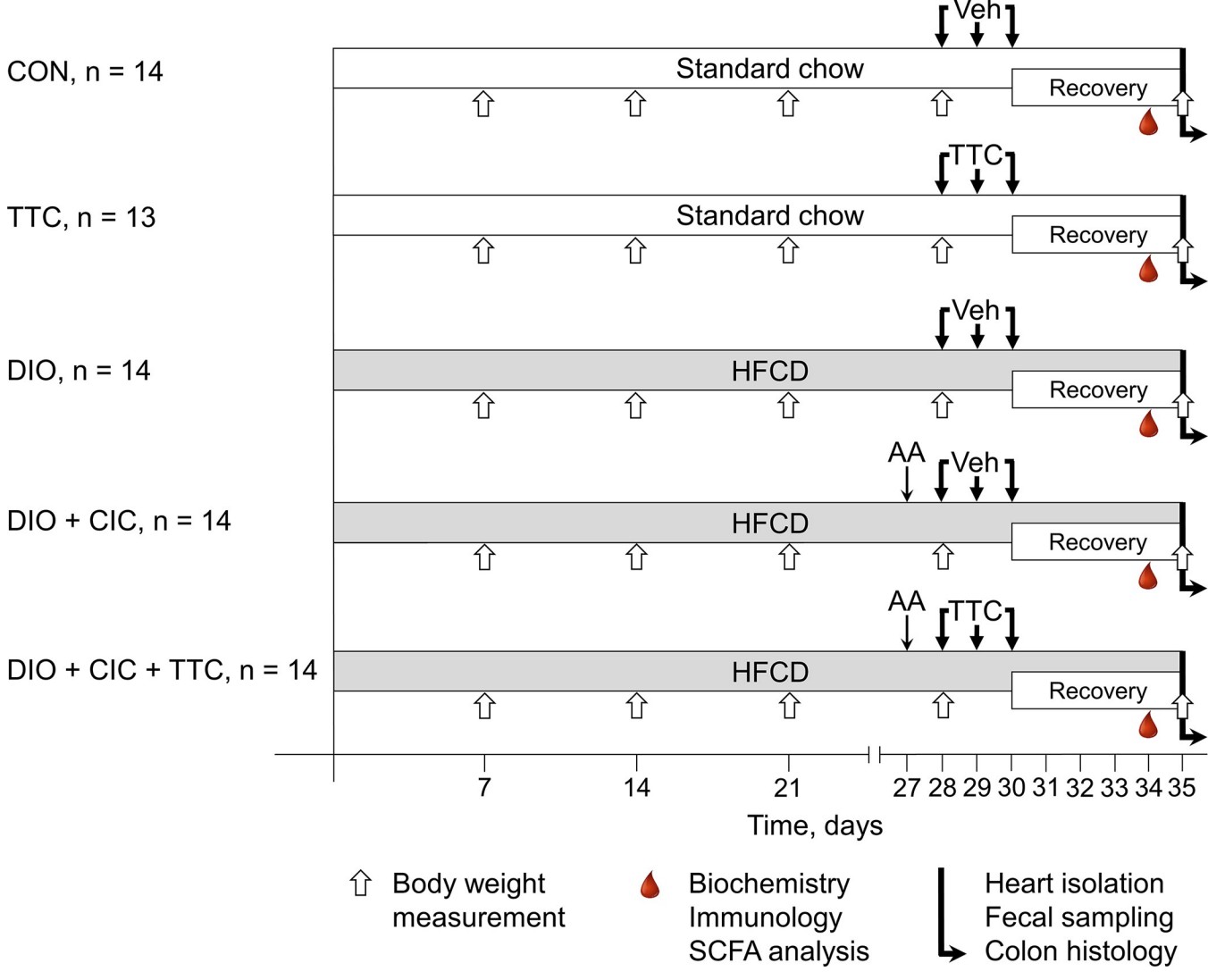

**Fig 1. Experimental design.** For details, see text. AA–acetic acid, HFCD–high-fat, high-carbohydrate diet, Veh–vehicle, TTC–tetracycline, SCFAs–short chain fatty acids.

normal saline), DIO (n = 14, rats with DIO were treated with the vehicle), DIO+CIC (n = 14, rats with DIO and CIC were treated with the vehicle), and DIO+CIC+TTC (n = 14, rats with DIO and CIC were treated with TTC). Each treatment was administered once daily for three days via the intragastric route. After the last day of treatment, the animals were allowed to recover for five days prior to heart isolation. All the animals, except those in the CON and TTC groups, were fed the HFCD during the recovery period. The CON and TTC groups received standard chow (12% fat, 0% sucrose; Lab Diet, St. Louis, MO, USA) during the study. At 9:00–10:00 am each day during the seven days prior to heart harvesting, the following details were recorded: clinical status, body weight, and water and food consumption. After autopsy was done, the cecum and the retroperitoneal, epididymal, and visceral fat pads were excised and weighed.

## Assessment of biochemical and immunological parameters

Saphenous vein blood was drawn for the analysis of biochemical and immunological parameters using a standard blood collection technique one day prior to heart isolation. Whole blood was centrifuged twice at 3000 rpm for 10 min to obtain serum. A biochemical analyzer (BioChem Analette; HTI, North Attleboro, MA, USA) was used to determine the serum levels of lactate, lactate dehydrogenase (LDH), alkaline phosphatase (ALP), urea, and uric acid. The serum concentrations of tumor necrosis factor-alpha (TNF-α), interleukin (IL)-8, monocyte chemoattractant protein-1 (MCP-1), C-reactive protein (CRP), and lipopolysaccharide (LPS) were measured using enzyme-linked immunosorbent assay (ELISA) kits (MR-96A; Mindray, Shenzhen, China) according to the manufacturer's instructions. Each ELISA test was performed in triplicate.

## DNA extraction and analysis of gut microbiota

Gut microbiota profiles in fecal samples were determined in two stages. First, DNA was extracted from the supernatants of fecal suspensions using a DNA extraction kit (QIAamp DNA Stool Mini Kit; QIAGEN, Hilden, Germany) in a dry block heater (Termit; DNA-Technology LLC, Moscow, Russian Federation), followed by incubation. Next, real-time polymerase chain reaction (PCR) was performed using T100™ Thermal Cycler (Bio-Rad, Hercules, CA, USA) and the reaction mixture Colonoflor-16 (Alpha-Lab, Saint-Petersburg, Russian Federation). Melt curve analysis was performed immediately after amplification to identify the targeted PCR product, which was then quantified using a spectrophotometer (NanoDrop ND-1000; Peqlab, Erlangen, Germany). The number of bacteria was expressed in colony-forming unit (CFU)/g.

## Analysis of SCFAs

Plasma concentrations of SCFAs were evaluated by gas chromatography with flame ionization detection (GC-FID). The concentrations of the following SCFAs were measured: acetic, propionic, isobutyric, and isovaleric acids. The SCFAs were identified by their retention times and characteristic mass fragment ions using the regimen of selected ion monitoring on a GC-FID system (Agilent 7890A; Agilent Technologies, Wilmington, NC, USA). SCFA quantification was done via automatic integration of chromatograms using GC/MSD ChemStation software (Agilent Technologies).

## Isolated heart perfusion according to Langendorff

The rats were anesthetized with sodium pentobarbital (60 mg/kg intraperitoneally). Each heart was excised via bilateral thoracotomy and perfused through the ascending aorta with Krebs-Henseleit buffer (consisting of the following [in mmol/L]: NaCl, 118.5; KCl, 4.7; $NaHCO_3$, 25; $KH_2PO_4$, 1.2; $MgSO_4$, 1.2; glucose, 11; and $CaCl_2$, 1.5) at a constant pressure of 85 mm Hg [17]. Perfusion pressure was maintained by gravity by using a water-jacketed double-walled glass column connected to the aortic cannula via a three-way stopcock. Buffer oxygenation was performed with carbogen (95% $O_2$ plus 5% $CO_2$) delivered through an inverted fritted glass filter.

Left ventricular systolic pressure (LVSP) and left ventricular end-diastolic pressure (LVEDP) were measured isovolumetrically using a nonelastic polyethylene balloon introduced into the left ventricle via the left atrium. The balloon was coupled to an insulin syringe and inflated with 0.4–0.6 mL of boiled water to obtain an LVEDP of < 10 mm Hg during stabilization. PhysExp Gold software (Cardioprotect Ltd., Saint Petersburg, Russian Federation) was

used to process the pressure wave recorded using a miniature pressure transducer (Baxter International, Deerfield, IL, USA). Values corresponding to the mean LVEDP and LVSP were obtained each minute. Left ventricular developed pressure (LVDP) was calculated as the difference between LVSP and LVEDP. Heart rate (HR) was derived from the pressure wave. Coronary flow rate (CFR) was determined by measuring the time for the collection of perfusate outflow. The core temperature of the hearts was maintained at 37˚C by water jacketing. The hearts of the rats were removed, cannulated, stabilized for 15 min, and then subjected to 30 min of global normothermic ischemia and 120 min of reperfusion. LVSP, LVEDP, HR, and CFR were measured 5 min before global ischemia and at the 15, 30, 45, 60, 75, 90, and 120th min of reperfusion. In addition, left ventricular pressure (LVP) was measured at the 5, 10, 15, 20, 25, and 30th min of global ischemia. Any heart with a HR of < 220 beats/min and a CFR > 18 or < 8 mL/min by the end of stabilization was excluded from the study. Hearts failing to show an LVDP > 100 mm Hg when LVEDP was maintained at < 10 mm Hg were also excluded from the study.

## Infarct size measurement

At the end of reperfusion, each heart was rapidly cut into four equally spaced transverse slices. The slices were immersed in a 1% solution of 2,3,5-triphenyltetrazolium chloride for 15 min at 37˚C. Stained slices were photographed with a stereomicroscope (SMZ18; Nikon, Tokyo, Japan) coupled to a digital camera (DS-Fi2, Nikon, Tokyo, Japan) for further examination of the non-stained (infarcted) areas. Infarct size have been delineated using software-based automatic discrimination of grey color gradations (ImageJ 1.34s; National Institutes of Health (NIH) Bethesda, MD, USA). The algorithm included the introduction of cutoff value of color intensity derived from the mean intensities typical of non-ischemic and necrotic tissue. Infarct size was expressed as a percentage of total ventricular area minus the cavities. The mean value for each section was used in the data analysis.

## Histopathological examination

Colon samples were fixed in buffered 10% paraformaldehyde, embedded in paraffin, cut into 5-μm sections, and stained with hematoxylin and eosin (H&E) for histological examination using standard techniques. After H&E staining, the slides were observed and photos were taken using an optical microscope (DM750; Leica, Wetzlar, Germany) at 100× magnification. The slides were analyzed by a pathologist who was blinded to the different treatments. We measured the thickness of the intestinal mucosal layer and the total thickness of the intestinal wall to verify CIC-associated inflammatory changes.

## Statistical analysis

All data are presented as mean ± standard deviation (SD). Statistical analysis was performed using SPSS 12.0 (IBM Corporation, Armonk, NY, USA). Differences in hemodynamic parameters over time in each group were analyzed using Friedman's repeated-measures analysis of variance (ANOVA) on ranks test followed by Dunn's multiple comparisons test. A post hoc test was performed only if the ANOVA analysis resulted in F < 0.05 and there was no variance in homogeneity. The Kruskal-Wallis test was used to determine differences in infarct size, number of gut microorganisms, SCFA level, biochemical and immunological parameters, and histological scores. This was followed by pairwise intergroup comparisons that were performed using nonparametric Mann-Whitney $U$ test. $P$ values < 0.05 were considered statistically significant.

## Results

### Body weight and consumption of food and water

At the end of the experiment, body weight was approximately 10% higher in the DIO group than in the CON group (405 ± 13 and 370 ± 10 g, respectively; $p < 0.05$; Fig 2A). Changes in body weight were not different between the CON and TTC groups; however, at day 35, body weight was significantly smaller in the DIO+CIC and DIO+CIC+TTC groups (337 ± 16 and 339 ± 14 g, respectively; $p < 0.05$) than in the CON group. Water intake was significantly higher in the DIO+CIC and DIO+CIC+TTC groups than in the CON group ($p < 0.05$, Fig

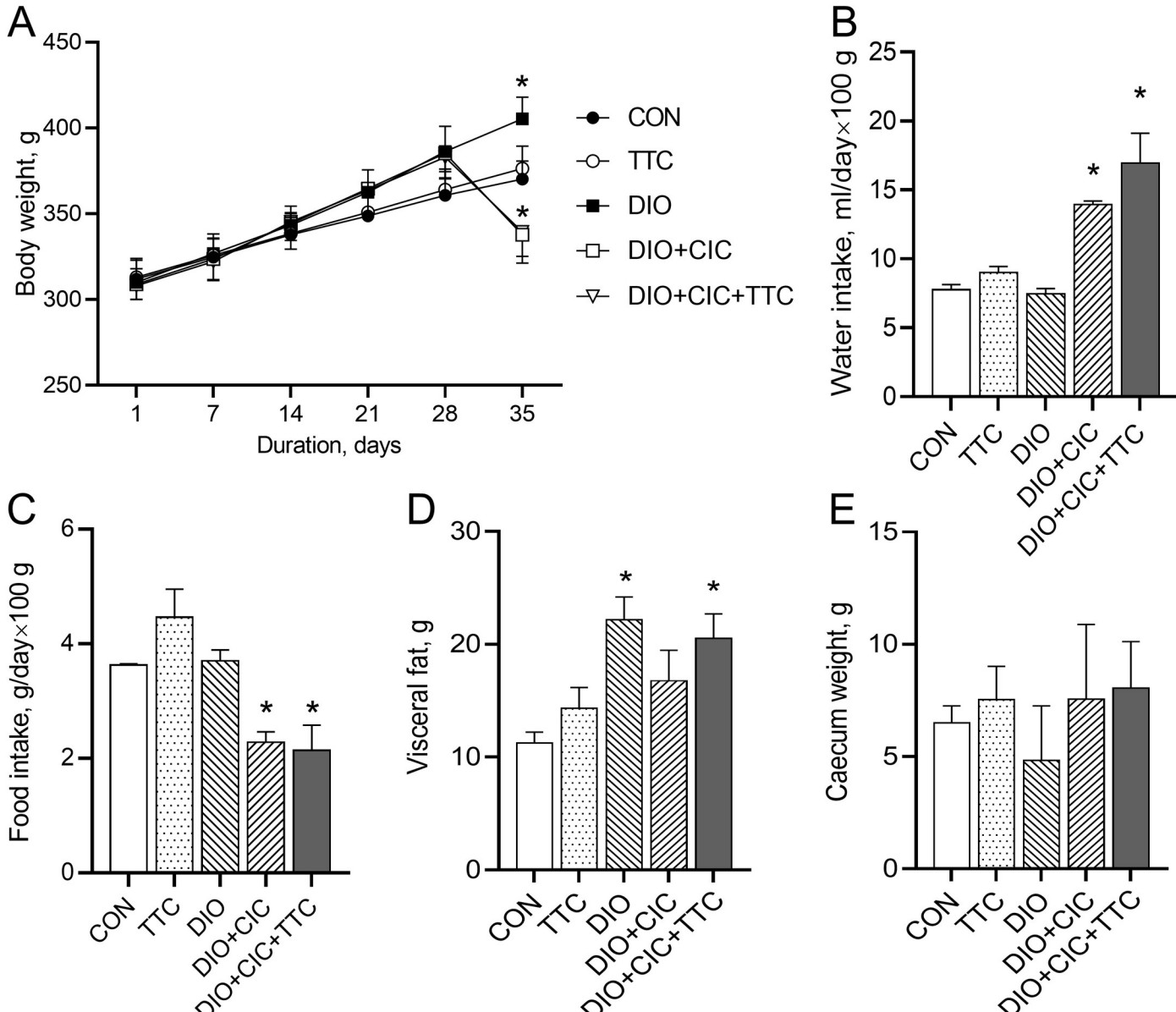

**Fig 2. Effects of different treatment regimens on animal body weight, food and water intake, and the weights of visceral fat and the cecum.** (A) Weekly changes in body weight, (B) water and (C) food intake, and (D) visceral fat and (E) cecum weights. Data are expressed as mean ± SD. * indicates $p < 0.05$ when data is compared to that for the CON group.

**Table 1. Biochemical serum markers in Wistar rats at the end of the experiment.** The results show mean values and standard deviations.

| | LAC (μM/L) | LDH (U/L) | ALP (U/L) | URE (μM/L) | URA (μM/L) |
|---|---|---|---|---|---|
| CON | 6.1 ± 2.4 | 456 ± 123 | 82 ± 34 | 4.4 ± 1.1 | 38.6 ± 9.8 |
| TTC | 10.4 ± 3.9 | 996 ± 196 | 218 ± 49* | 7.6 ± 0.7* | 79.6 ± 10.9* |
| DIO | 7.7 ± 3.3 | 503 ± 218 | 106 ± 29 | 4.9 ± 1.0 | 42.5 ± 3.5 |
| DIO+CIC | 5.9 ± 0.9 | 1269 ± 432 | 151 ± 80 | 4.1 ± 0.7 | 71.9 ± 20.9 |
| DIO+CIC+TTC | 33.7 ± 3.9* | 1640 ± 550* | 415 ± 121* | 7.8 ± 0.9* | 273 ± 77* |

*—$p < 0.05$ versus controls. LAC–lactate, LDH–lactate dehydrogenase, ALP–alkaline phosphatase, URE–urea, URA–uric acid.

2B). Food intake was significantly lower in the DIO+CIC and DIO+CIC+TTC groups than in the CON group ($p < 0.05$, Fig 2C). Visceral fat weight was significantly higher in the DIO and DIO+CIC+TTC groups than in the CON group ($p < 0.05$, Fig 2D). There were no differences in cecum weight between the groups (Fig 2E).

## Biochemical and immunological parameters

TTC significantly elevated the serum levels of ALP, urea, and uric acid in the healthy animals (Table 1). Serum biochemical parameters were not different in the DIO, DIO+CIC, and CON groups. Furthermore, LDH, ALP, urea, and uric acid levels were higher in the DIO+CIC+TTC group than they were in the CON group. Additionally, the levels of proinflammatory cytokines, CRP, and LPS remained unchanged in the TTC group (Table 2). LPS level was significantly higher in the DIO group than in the CON group, which might be attributed to the effect of metabolic endotoxemia. The DIO+CIC group was characterized by elevated serum concentrations of TNF-α, IL-8, MCP-1, and LPS. However, TTC significantly reduced the levels of IL-8 and MCP-1 in the animals with both DIO and CIC. The serum concentrations of TNF-α and LPS were elevated in the DIO+CIC+TTC group as they were in the DIO+CIC group.

## Intestinal microbiome

The relative numbers of some intestinal bacteria (*Lactobacillus spp.*, *Bifidobacterium spp.*, *Escherichia coli*, *Faecalibacterium prausnitzii*, *Bacteroides fragilis*, and *Akkermansia mucini-phila*), as well as the total bacterial count, are presented in Fig 3. Total bacterial count and *Bacteroides fragilis* count were significantly lower in the TTC and DIO+CIC+TTC groups than in the CON group (Fig 3A and 3F). Additionally, *Lactobacillus spp.*, *Bifidobacterium spp.*, and

**Table 2. Serum levels of proinflammatory cytokines, C-reactive protein, and lipopolysaccharide in Wistar rats at the end of the experiment.** The results show mean values ± SD.

| | TNF-α (pg/ml) | IL-8 (pg/ml) | MCP-1 (pg/ml) | CRP (pg/ml) | LPS (ng/ml) |
|---|---|---|---|---|---|
| CON | 3.4 ± 1.5 | 3.4 ± 0.5 | 58.6 ± 2.4 | 1.2 ± 0.4 | 6.2 ± 2.8 |
| TTC | 3.1 ± 1.9 | 3.1 ± 1.0 | 45.8 ± 17.4 | 1.1 ± 0.4 | 11.1 ± 3.2 |
| DIO | 5.5 ± 1.9 | 6.2 ± 1.7 | 52.4 ± 4.8 | 1.5 ± 0.2 | 13.3 ± 1.4* |
| DIO+CIC | 9.7 ± 2.7* | 12.5 ± 3.9* | 260 ± 91* | 1.6 ± 0.5 | 18.4 ± 7.2* |
| DIO+CIC+TTC | 10.5 ± 2.6* | 4.8 ± 1.6# | 157 ± 48# | 1.2 ± 0.2 | 16.2 ± 5.6* |

*–$p < 0.05$ versus controls

#–$p < 0.05$ versus DIO + CIC. TNF-α –tumor necrosis factor-alpha, IL-8 –interleukin-8, MCP-1 –monocyte chemoattractant protein-1, CRP–C-reactive protein, LPS–lipopolysaccharide.

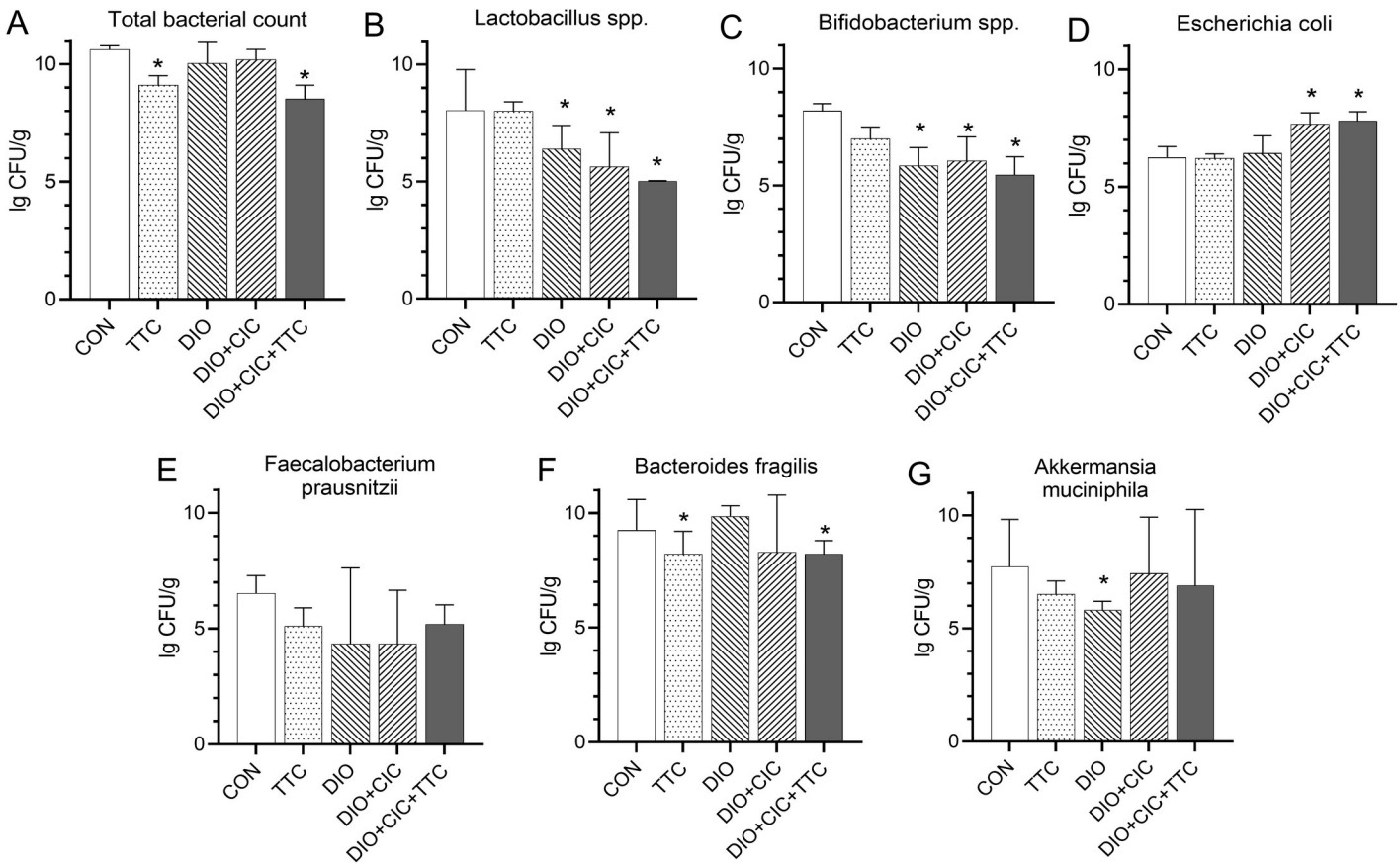

**Fig 3. Effects of TTC on intestinal microbiota composition in healthy rats and obese rats with CIC.** (A) Total bacterial count. (B) *Lactobacillus spp.*, (C) *Bifidobacterium spp.*, (D) *E. coli*, (E) *Faecalibacterium prausnitzii*, (F) *Bacteroides fragilis*, and (G) *Akkermansia muciniphila* counts in fecal samples were analyzed by RT-PCR. Results are presented as mean ± SD and expressed in CFU/g. * indicates $p < 0.05$ when data is compared to that for the CON group.

*Akkermansia muciniphila* counts were found to be reduced in the DIO group (Fig 3B, 3C and 3G). The animals with both DIO and CIC had lower counts of *Lactobacillus spp.* and *Bifidobacterium spp.* but a higher count of *Escherichia coli* than those in the CON group did (Fig 3B–3D). Similar changes in microbiota profiles were observed in the DIO+CIC+TTC group. However, the population of *Faecalibacterium prausnitzii* was not different among the five groups.

### Short-chain fatty acids

The serum levels of the four SCFAs at the end of the experiment are shown in Fig 4. Serum acetate level was significantly higher in the TTC, DIO+CIC+TTC, and DIO+CIC groups (11.6 ± 3.1, 13.6 ± 2.9, and 10.6 ± 2.9 mmol/L, respectively; $p < 0.0001$) than in the CON group (0.7 ± 0.1 mmol/L) (Fig 4A). DIO alone was not associated with an elevated acetate level (1.4 ± 0.6 mmol/L, $p > 0.05$ when the value is compared to the data for the CON group). The levels of propionic, isobutyric, and isovaleric acids were not different among the groups (Fig 4B–4D).

### Isolated heart function and myocardial infarct size

Changes in LVP that occurred during the global ischemia are shown in Fig 5A. Ischemic contracture was defined as follows: at least a threefold increase in LVP at any time during the

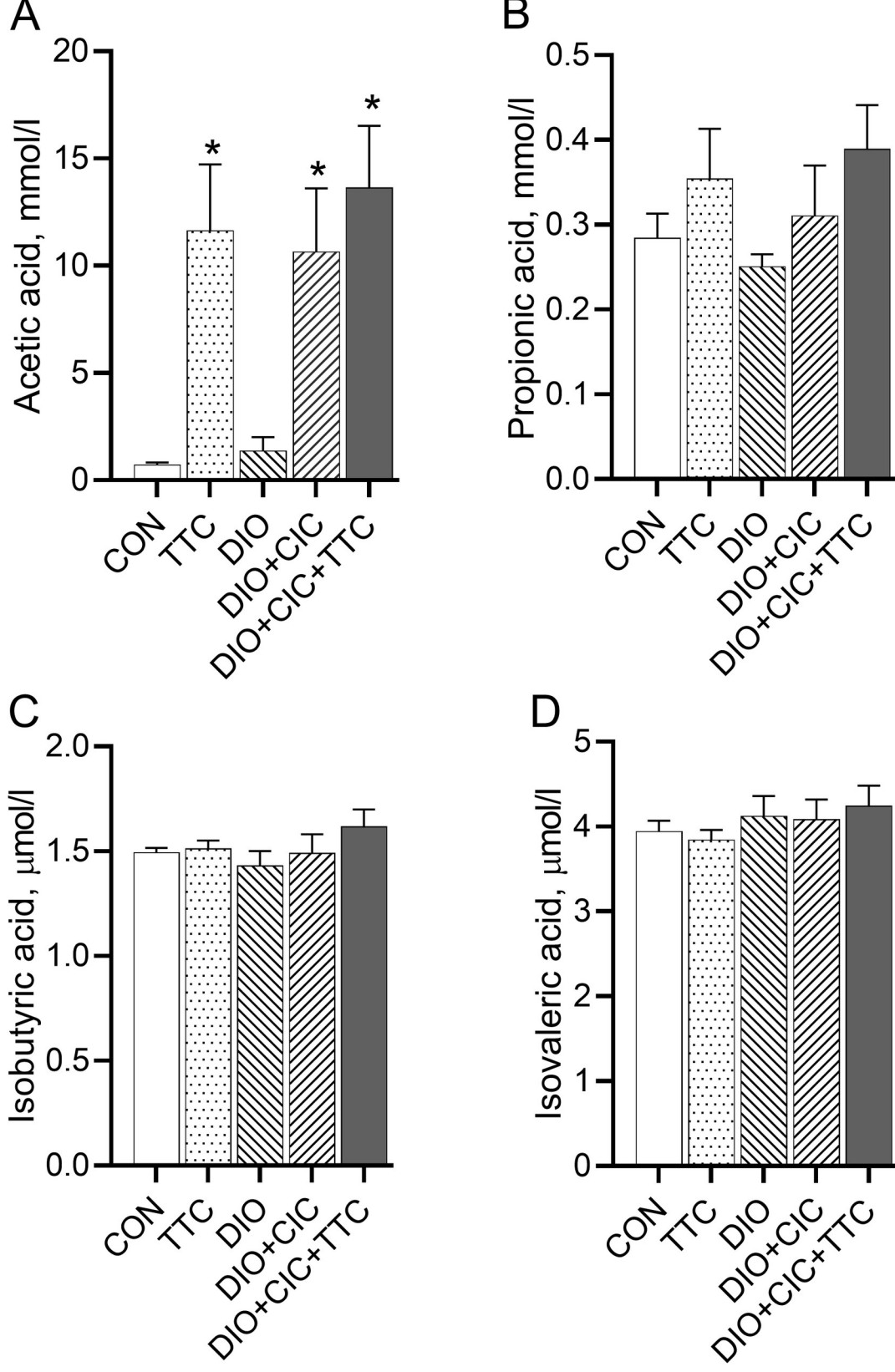

**Fig 4. Effects of TTC on plasma SCFA levels in healthy rats and obese rats with CIC.** Plasma concentrations of (A) acetic acid, (B) propionic acid, (C) isobutyric acid, and (D) isovaleric acid were determined by GC-FID. Results are presented as mean ± SD. * indicates $p < 0.05$ when data is compared to that for the CON group.

ischemic period relative to the LVP after five minutes of ischemia. The LVP during the initial twenty minutes of ischemia tended to be higher in the DIO and DIO+CIC groups. The baseline LVDP, LVEDP, and CFR values were similar in all the groups. No important intergroup differences in functional parameters during the reperfusion period were found (Fig 5B–5D). The HR data are provided in S1 Table.

Myocardial infarct size was 62 ± 4% in the CON group (Fig 5E). TTC significantly limited infarct size in the healthy animals (50 ± 7%, $p < 0.05$ when the value is compared to the data for the CON group). Infarct size was not significantly different between the CON, DIO, and DIO+CIC groups (62 ± 4, 61 ± 3, and 67 ± 6%, respectively). Moreover, infarct size was significantly larger in the DIO+CIC+TTC group than in the CON group (77 ± 5%, $p < 0.05$).

## Histopathological findings

The histological structure of the colon was normal in the CON, TTC, and DIO groups (Fig 6A–6C). However, colon samples from the DIO+CIC and DIO+CIC+TTC groups had areas with marked ulcer and erosive lesions filled with purulent exudate. Crypt necrosis, mucosal edema, and massive polymorphonuclear leukocyte infiltration were also evident from the microscopic analysis (Fig 6D and 6E). In addition, the samples had increased numbers of goblet cells in the colonic mucosa. Severe inflammatory changes in the colon resulted in a significant increase in total intestinal wall thickness in the DIO+CIC and DIO+CIC+TTC groups (Fig 6F). Mucosal membrane thickness tended to be smaller in the groups with colitis (Fig 6F).

## Discussion

One of the main findings of the present study was that the intragastric administration of 15 mg TTC for three days resulted in infarct size limitation in the isolated rat heart subjected to global ischemia-reperfusion. The comorbidity of DIO and CIC resulted in the reversal of the effects of TTC, as was evidenced by infarct size being significantly larger in the DIO+CIC+TTC group than in the CON group.

To the best of our knowledge, there is only one published study on the cardioprotective effect of TTC [9]. In that work, TTC (at a dose of 4 mg/kg) caused infarct size limitation in an *in vivo* murine model of myocardial infarction when it was administered 30 min prior to coronary occlusion. TTC elicited an anti-stunning effect at the same dose in a dog model of reversible myocardial ischemia. The results of the present study confirm previous findings that TTC alleviates myocardial IRI, now in the *ex vivo* model of myocardial IRI. Although we observed infarct size limitation in TTC group, this was not associated with significant improvement in myocardial function. According to our experience, myocardial infarct size and functional performance of the isolated heart subjected to ischemia-reperfusion are not always showing parallel changes, especially when the extent of cardioprotection is not very strong. It was suggested that the cardioprotective effect of TTC could be attributed to the direct effect of TTC on the heart, particularly resulting in augmented expression of stress proteins due to partial inhibition of protein translation in mitochondria. TTC increased the expression of profilin 1, cyclophilin A, transglutaminase II, and heat shock proteins 27 and 70 in HeLa cells and the myocardium. Studies on the chemical and pharmacokinetic properties of minocycline and doxycycline, which are also tetracyclines, have revealed that these substances may have a slightly greater cardioprotective potency than TTC has. It is reported that minocycline significantly reduces

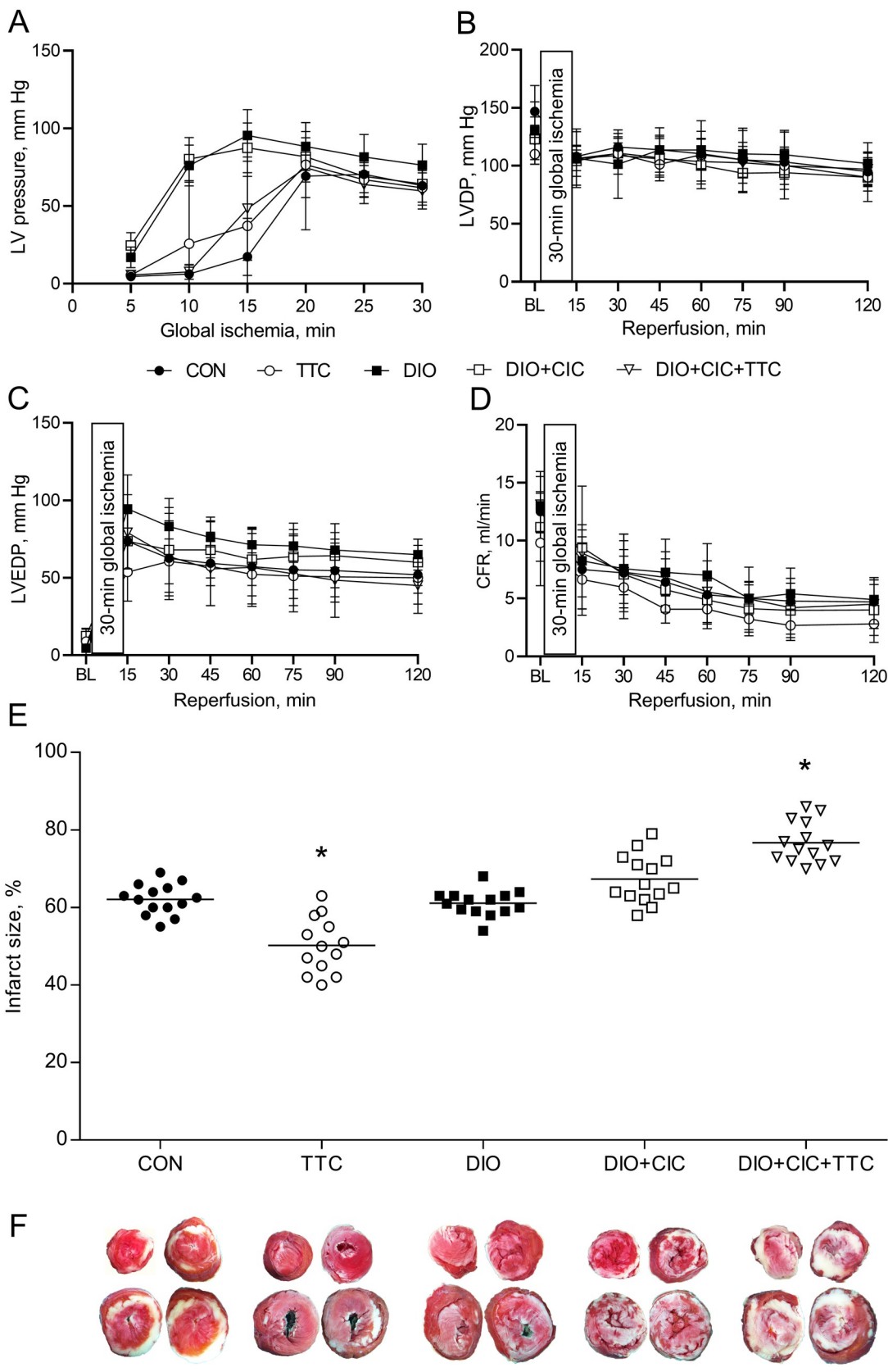

**Fig 5. Functional parameters and myocardial infarct size in isolated rat hearts subjected to 30 min of global ischemia followed by 120 min of reperfusion.** (A) Ischemic contracture, (B) LVP, (C) LVEDP, and (D) CFR values at baseline and during reperfusion. Data are presented as mean ± SD. (E) Infarct size results are presented as dot plots with median values. (F) Representative images of heart slices stained with triphenyltetrazolium chloride. * indicates $p < 0.05$ when data is compared to that for the CON group.

infarct size in rats when it is administered before and after ischemia [18], which might be due to the antioxidant activity of minocycline. It has also been shown that minocycline and vitamin E have similar free radical-scavenging properties [19]. In addition, minocycline can alleviate the consequences of oxidative injury by inhibiting inducible nitric oxide synthase [20] and poly (ADP-ribose) polymerase 1 [21]. Minocycline has a strong anti-apoptotic effect because it inhibits the activity of caspases. It also decreases the expression and release of the proapoptotic protein Smac/DIABLO from mitochondria into the sarcoplasm [22]. Doxycycline inhibits the activity of matrix metalloproteinases more than TTC or minocycline does due to the high chelating capacity of doxycycline with respect to $Zn^{2+}$. This effect is relevant in cardioprotection because ischemia-reperfusion in the rat heart results in an increased activity of matrix metalloproteinase-2 [23]. However, doxycycline prevents this activation, improves functional recovery during reperfusion, and decreases the proteolysis of troponin I and myosin. This means that tetracyclines exhibit a protective effect against IRI to different extents.

Recent evidence suggests that the cardioprotective effect of some antimicrobial agents possibly arises from changes in the composition of intestinal microbiota with resulting systemic neuroimmunoendocrine shifts rather than from a direct effect on the myocardium. The first report on the effect of gut microbiome composition on myocardial infarct size in rats was published in 2012 [7]. It was noted in the study that vancomycin or *L. plantarum 299V* decreased plasma leptin level and infarct size by 27 and 29%, respectively. The same group later demonstrated that pretreatment of rats with vancomycin or a mixture of streptomycin, neomycin, bacitracin, and polymyxin B resulted in infarct size reduction and associated changes in gut microbiota [8]. Furthermore, metabolomic analysis of plasma showed 284 differentially expressed metabolites, 193 of which were decreased in the animals treated with antibiotics. However, treatment with some exogenous amino acid metabolites that were negatively expressed resulted in inhibition of vancomycin-mediated protection of the myocardium. In the present study, we were interested in exploring the relationships between TTC-induced cardioprotection and gut microbiome changes, plasma SCFA levels, and proinflammatory cytokine levels. Moreover, we analyzed the effects of TTC on myocardial IRI in both healthy rats and obese rats with CIC. This is crucial for overcoming existing translational barriers such as failure of cardioprotective therapies in clinical trials despite very promising results in preclinical studies [24]. Comorbidity is one of the main factors responsible for the low clinical efficacies of infarct-limiting interventions [25]. For example, the protective effects of ischemic pre- and postconditioning are inhibited in animals with hypercholesterolemia [26], diabetes mellitus, and obesity [27]. In this regard, it is becoming increasingly important to test cardioprotective interventions in aged animals with clinically relevant comorbidities [28]. In the present study, well established and validated rat models of HFCD-induced obesity and acetic acid-induced acute colitis were used. DIO development was verified based on body weight gain and a significant increase in visceral fat weight.

The effects of DIO on myocardial tolerance to IRI remain controversial. It has been indicated that cardiac IRI is aggravated in animals with high fat diet (HFD)-induced obesity. This is consistent with clinical data on the increased susceptibility of obese patients to myocardial ischemia [29]. In contrast, other reports have suggested that HFD can cause paradoxical increase in cardiac resistance to IRI, which might be accounted for by delayed normalization

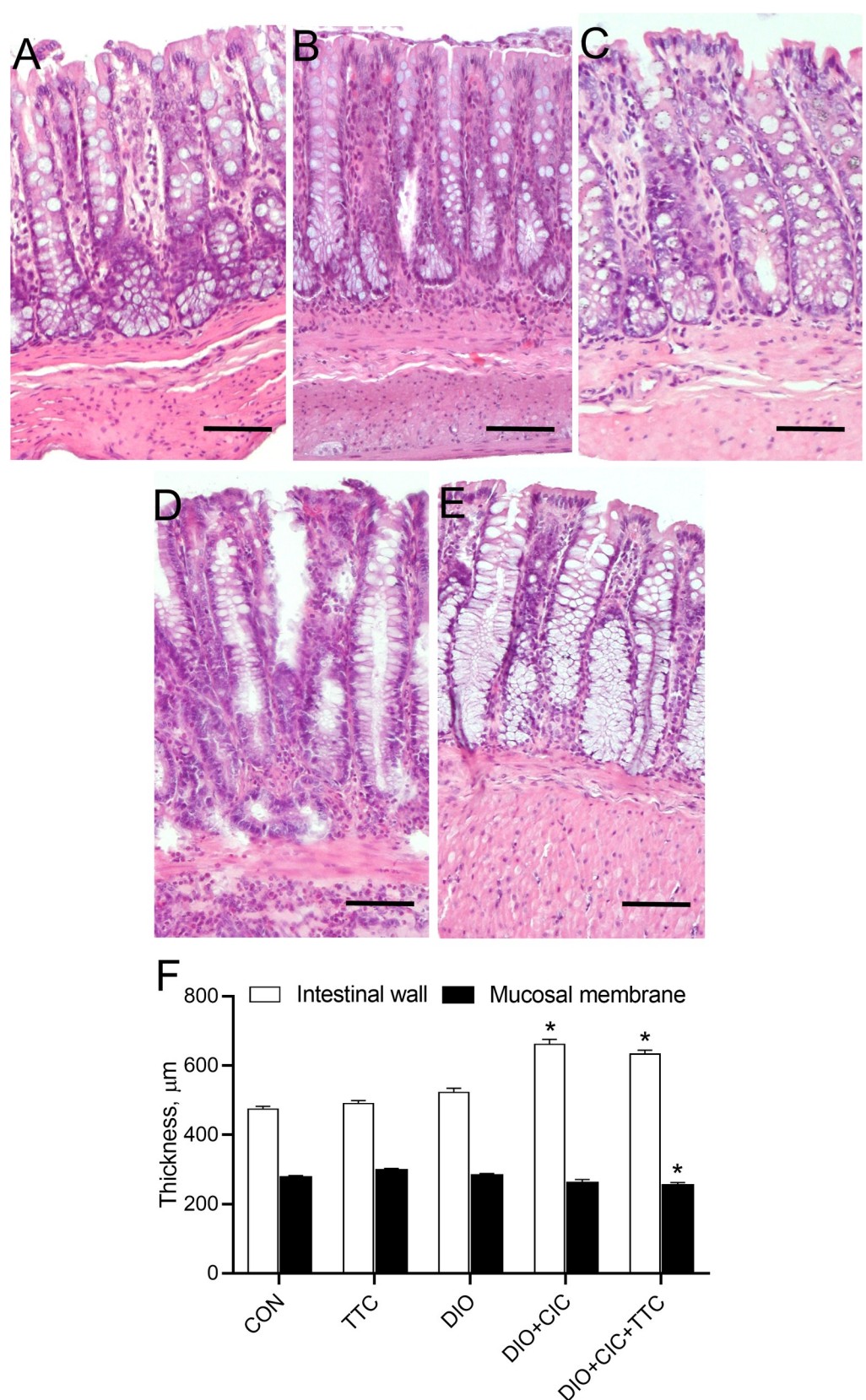

**Fig 6. Validation of CIC model via histopathological examination of the colon.** Colon samples were collected from animals in the (A) CON, (B) TTC, (C) DIO, (D) DIO+CIC, and (E) DIO+CIC+TTC groups at the end of the experiment, fixed in paraformaldehyde, and stained with H&E. Morphometric analysis was performed to determine (F) total intestinal wall thickness and mucosal membrane thickness. Tissue sections were observed under a microscope at 100× magnification. The scale bar is 100 μm for all images. The images are representative of at least four independent sections. * indicates $p < 0.05$ when data is compared to that for the CON group.

of intracellular pH upon reperfusion [30,31]. Since we have not studied myocardial tolerance to IRI in the *in vivo* model in the present study, we can only conclude that DIO in this study had no effect on myocardial tolerance to IRI in the isolated heart model.

CIC development in the present study was confirmed using histological findings. Additionally, elevated levels of proinflammatory cytokines and the occurrence of functional disorders such as weight loss, decreased food intake, and increased water intake indicated that CIC had been successfully developed. TTC did not induce any specific changes in microbiome or cytokine levels in the healthy animals, which might be due to the global spread of TTC-resistant bacteria. The only important finding in TTC group was dramatic elevation in plasma acetate concentration, which could be equally attributed to gut microbiota and tissues. A similar increase in plasma acetate level was observed in the TTC and DIO+CIC+TTC groups, which were characterized by the smallest and largest infarct sizes, respectively. This data suggests that there is no association between TTC-mediated cardiac protection and plasma SCFA levels, which indicates that TTC protects the heart mainly by having a direct effect on cardiac cells. There are, however, dozens of other metabolites that were not analyzed in the present study. In a previous study, significant changes in the plasma levels of 22 metabolites in TTC-treated rats were observed. Additionally, a highly specific decrease in 3-indoxylsulfate concentration was noted [32]. The functional significance of these changes in relation to infarct size limitation, as well as the potential involvement of free fatty acid receptor 2/3 (FFAR2/3)-mediated signaling in cardioprotection against IRI remain to be explored. There is no information yet on the effects of FFAR2/3 activation in the heart; however, the effects of FFAR2/3 activation in the pancreas and fat tissues have been extensively studied [33].

The present study revealed some important changes in the TTC and DIO+CIC+TTC groups that might have been responsible for the loss of the protective effect of TTC in the DIO+CIC+TTC group. Changes in intestinal microbiota composition as well as increased LPS and proinflammatory cytokine levels were noted in the rats with both DIO and CIC. It was observed that the comorbidity of DIO and CIC was associated with decreased counts of *Lactobacillus spp.* and *Bifidobacterium spp.* with a concomitant increase in *E. coli* count, which is typical of these disorders [34]. Recent study in aged mice showed that feeding them with high-calorie obesogenic diet resulted in significant changes in intestinal microbiome with marked expansion of the genus *Allobaculum* in the fecal microbiota [35]. The differing pattern of microbiome between healthy animals and those with comorbidity allows us to hypothesize that the extent of intestinal dysbiosis may correlate with infarct size. The lack of standardization of dysbiosis severity hinders quantitative estimation of this correlation. Currently, there is no data on an association between gut microbiota profiles and reduced myocardial tolerance to IRI. Our data shows that specific changes in the gut microbiome might interfere with some cardioprotective therapies. Furthermore, plasma LPS level was found to be significantly increased in the DIO, DIO+CIC, and DIO+CIC+TTC groups. For the DIO group, this increase in LPS level could be due to metabolic endotoxemia [36]. However, the mechanism underlying the elevated LPS level in DIO+CIC group seems to be more complex, involving both metabolic endotoxemia and marked increase in gut barrier permeability as a result of severe inflammation [37]. It is well established that low doses of LPS can alleviate myocardial

IRI due to activation of the inositol-requiring enzyme 1 and phosphoinositide 3-kinase/Akt signaling pathways [38,39]. However, high doses of LPS aggravate IRI and cause contractile dysfunction [40]. We presume that the significantly elevated LPS level in the DIO+CIC group resulted in a subthreshold injury to the heart, which neutralized the protective effect of TTC. The above-mentioned dual effect of LPS on the heart is analogous to that of cytokines. It is reported that myocardial tolerance to IRI in mice could be increased after the animals are treated with cytokines such as TNF-α and interleukin-1β [41,42]. However, when administered at high doses, TNF-α inhibits myocardial contractility and augments the release of myoglobin in the isolated rat heart [43]. In the present study, threefold increases in TNF-α and MCP-1 levels in the animals with CIC resulted in inhibition of TTC-induced cardioprotection.

The present study had several methodological limitations. Firstly, RT-PCR, instead of the "gold standard" metagenomic analysis, was used for microbiome characterization. This was because at the current stage of our project we intended to perform targeted identification of certain bacteria genera and species. Secondly, TTC-mediated cardioprotection was not assessed separately in the DIO and CIC groups but only in the animals with both pathologies. Thirdly, the molecular mechanism(s) underlying TTC-induced cardioprotection and its loss in the rats with both DIO and CIC were not investigated. Presently, we can only discuss the associations between infarct size, microbiota composition, and plasma cytokine levels. However, the causal roles of these factors in infarct size limitation should be investigated in additional proof-of-concept studies.

In conclusion, our data confirms that TTC has a cardioprotective effect in healthy animals. However, this effect was reversed in obese animals with CIC, which was associated with specific changes in gut microbiota and significantly elevated levels of cytokines and LPS in plasma. Our findings indicate that certain antimicrobial drugs should not be used in patients with obesity and coexisting inflammatory diseases. This is because of the increased risk of myocardial injury and an unfavorable outcome of myocardial infarction.

## Supporting information

**S1 Table. The values of heart rate (beats/min) in the experimental groups.** Data are mean ± SD. Group legends: controls (CON), tetracycline-treated controls (TTC), diet-induced obesity (DIO), diet-induced obesity plus chemically induced colitis (DIO + CIC), tetracycline-treated rats with diet-induced obesity plus chemically induced colitis (DIO + CIC + TTC). (DOCX)

## Author Contributions

**Conceptualization:** Yury Yu Borshchev, Michael M. Galagudza.

**Data curation:** Yury Yu Borshchev, Sarkis M. Minasian, Inessa Yu Burovenko, Victor Yu Borshchev, Olga V. Borshcheva.

**Formal analysis:** Yury Yu Borshchev, Inessa Yu Burovenko, Victor Yu Borshchev, Natalia Yu Semenova, Olga V. Borshcheva.

**Funding acquisition:** Michael M. Galagudza.

**Investigation:** Yury Yu Borshchev, Sarkis M. Minasian, Egor S. Protsak, Olga V. Borshcheva.

**Methodology:** Yury Yu Borshchev, Sarkis M. Minasian, Inessa Yu Burovenko, Egor S. Protsak, Natalia Yu Semenova.

**Project administration:** Yury Yu Borshchev, Michael M. Galagudza.

**Software:** Inessa Yu Burovenko, Natalia Yu Semenova.

**Validation:** Sarkis M. Minasian, Egor S. Protsak.

**Visualization:** Victor Yu Borshchev, Natalia Yu Semenova.

**Writing – original draft:** Yury Yu Borshchev, Michael M. Galagudza.

**Writing – review & editing:** Michael M. Galagudza.

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
