## [Decision Letter · Decision Letter 0]

17 Sep 2019

PONE-D-19-22988

Effects of tetracycline on myocardial infarct size in obese rats with chemically-induced colitis

PLOS ONE

Dear Dr. Galagudza,

Thank you for submitting your manuscript to PLOS ONE. After careful consideration, we feel that it has merit but does not fully meet PLOS ONE’s publication criteria as it currently stands. Therefore, we invite you to submit a revised version of the manuscript that addresses the points raised during the review process.

We would appreciate receiving your revised manuscript by Nov 01 2019 11:59PM. To enhance the reproducibility of your results, we recommend that if applicable you deposit your laboratory protocols in protocols.io, where a protocol can be assigned its own identifier (DOI) such that it can be cited independently in the future. For instructions see: http://journals.plos.org/plosone/s/submission-guidelines#loc-laboratory-protocols

We look forward to receiving your revised manuscript.

Kind regards,

Michael Bader

Academic Editor

PLOS ONE

Journal Requirements:

2. In your Discussion, please discuss the possible limitations of the study,  including the need for clinical studies before any conclusion regarding human health can be made.

3. To comply with PLOS ONE submissions requirements, in your Methods section, please provide additional information on the animal research and ensure you have included details on (1) methods of sacrifice, (2) methods of anesthesia and/or analgesia, and (3) efforts to alleviate suffering.

Reviewers' comments:

Reviewer's Responses to Questions

**Comments to the Author**

1. Is the manuscript technically sound, and do the data support the conclusions?

Reviewer #1: Partly

Reviewer #2: Partly

2. Has the statistical analysis been performed appropriately and rigorously? 

Reviewer #1: No

Reviewer #2: Yes

3. Have the authors made all data underlying the findings in their manuscript fully available?

Reviewer #1: Yes

Reviewer #2: Yes

4. Is the manuscript presented in an intelligible fashion and written in standard English?

Reviewer #1: Yes

Reviewer #2: Yes

5. Review Comments to the Author

Reviewer #1: Borshchev and colleagues determined the effects of tetracycline (TTC) on myocardial infarct size in obese rats following chemically-induced colitis (CIC). To accomplish this, authors used a single dose of rectal administration of 3 % acetic acid to the rats that induced CIC. Overall interesting manuscript, needs some additional clarification and precise interpretation of infarction area and microbiome results.

1. Add detailed information of age and sex of rats that were used in the presented study.

2. Infarct size is variable outcome depending on the color density therefore need additional clarification on this aspect.

3. Most important, the infarct size not infarct area is measured and measured only ex-vivo hearts, instead of in-vivo.

4. Does the provided ex-vivo changes align with in-vivo changes in the literature or any previosu results, please discuss.

5. Elaborate microbiome results, in contest of inflammation and possible dysregulation.

6. Measured short chain fatty acids are determined in serum, however in cardiac tissue will be of interest to determine the impact on infarct size/area.

7. Provide rationale for short chain fatty acids measurements, in context of micriobiome.

8. For microbiome data integration with cardiac outcome refer particularly extensive statistical analyses of intestinal microbime data - PMCID: PMC6463911

9. This multilayered study needs clarified presentation which lacking in the entire manuscript.

10. In discussion suggest, how obesity impacted the overall cardiac dysregulation of metabolism in-vivo tissue rather than ex-vivo.

Reviewer #2: In their manuscript, Borshchev et al. investigated the effect of the gut microbiota on the extent of myocardial injury after ischemia. They observed a potentially beneficial effect of tetracycline (TTC)-mediated alteration of distinct intestinal bacteria in rats fed a standard diet whereas these effects were abolished in high fat diet induced obesity and chemically induced colitis. Overall, the novelty of the current study is limited since cardioprotective effects of TTC in an experimental myocardial infarction model has been previously postulated. Moreover, it remains unclear why the authors treated chemically induced colitis to investigate TTC-related effects on ischemic myocardial injury.

Further major points that need to be clarified are as follows:

1. Please explain why despite differences in the extent of myocardial injury upon treatment with tetracycline, the hemodynamic parameters are not affected.

2. What are putative underlying mechanisms of TTC-mediated cardiac protection in lean mice and which mechanisms are dysregulated leading to detrimental effects of TTC in chemical induced colitis.

3. Measuring SCFAs in the plasma is not expedient, since most of the SCFAs are metabolized in the liver before reaching the circulation. Thus, to assess changed in SCFA production, it is recommended to measure SCFAs in the feces. Protective effects of SCFAs are considered to be based on local paracrine mechanisms in the intestine (such as intestinal immune homeostasis) with secondary systemic consequences.

6. PLOS authors have the option to publish the peer review history of their article (what does this mean?). If published, this will include your full peer review and any attached files.

Reviewer #1: No

Reviewer #2: No

---

## [Author Response · Author response to Decision Letter 0]

2 Oct 2019

Response to the comments by Reviewer 1

We would like to thank the reviewer for the constructive comments on the manuscript.

1. Critique: “Add detailed information of age and sex of rats that were used in the presented study.” 

Response: Male outbred Wistar rats (age, 11–12 weeks; weight, 300–320 g) were used throughout the experiments. The details on animals’ age and sex have been added (p. 5, “Animals” section).

2. Critique: “Infarct size is variable outcome depending on the color density therefore need additional clarification on this aspect.”

Response: Infarct size have been delineated using software-based automatic discrimination of grey color gradation. The algorithm included the introduction of cutoff value of color intensity derived from the mean intensities typical of non-ischemic and necrotic tissue. Relevant considerations have been included in the “Infarct size measurement” section (p. 10, first paragraph).

3. Critique: “Most important, the infarct size not infarct area is measured and measured only ex-vivo hearts, instead of in-vivo.”

Response: We apologize for imprecise interpretation of the terms. Indeed, the infarct size has been measured only in the ex vivo hearts subjected to global ischemia-reperfusion in this study. Therefore, we carefully checked the manuscript and corrected the terms in the Abstract and in the Introduction sections (e.g. “The aim of this study was to … on myocardial infarct size in the isolated hearts obtained from obese rats with …”, “we investigated the effects of TTC on myocardial IRI in the isolated hearts of rats with …”).

4. Critique: “Does the provided ex-vivo changes align with in-vivo changes in the literature or any previous results, please discuss.”

Response: Yes, the infarct-limiting effect of tetracycline has been previously demonstrated in the in vivo murine model of myocardial infarction (Kagawa N. et al., 2005). Our data fit well with these findings, but now in the ex vivo model. We discussed this on p. 17, Discussion. In addition, we plan to perform next study in the in vivo rat model of infarction in order to compare the results of these two models.

5. Critique: “Elaborate microbiome results, in context of inflammation and possible dysregulation.”

Response: We observed significant reduction in Lactobacillus spp., Bifidobacterium spp., and Akkermansia muciniphila counts in diet-induced obesity. The animals with both obesity and colitis had lower counts of Lactobacillus spp. and Bifidobacterium spp. but a higher count of Escherichia coli than in controls. Healthy TTC-treated rats demonstrated significant reduction in total bacterial count without appreciable changes in particular species except Bacteroides fragilis. This differing pattern of microbiome between healthy and comorbid animals allows us to hypothesize that the extent of dysbiosis may correlate with infarct size. The lack of standardization of dysbiosis severity hinders quantitative estimation of this correlation. A more detailed analysis of microbiota composition may contribute to the understanding of this issue. Please see changes on p. 21.

6. Critique: “Measured short chain fatty acids are determined in serum, however in cardiac tissue will be of interest to determine the impact on infarct size/area.”

Response: In this study, we were interested to analyze the serum levels of short chain fatty acids (SCFAs) as potential preconditioning factors inducing myocardial tolerance to ischemia-reperfusion injury. Since the data on local myocardial production of SCFAs are currently lacking, we presumed that serum concentration of SCFAs reflects well their myocardial tissue content.

7. Critique: “Provide rationale for short chain fatty acids measurements, in context of microbiome.”

Response: It is well established that short chain fatty acids are produced by the intestinal microbiota. In this study, we were interested to analyze the serum levels of short chain fatty acids (SCFAs) as potential preconditioning factors inducing myocardial tolerance to ischemia-reperfusion injury. However, due to limited number of bacterial species analyzed, in the present study we cannot determine the exact species responsible for elevation of acetate in some groups.

8. Critique: “For microbiome data integration with cardiac outcome refer particularly extensive statistical analyses of intestinal microbiome data - PMCID: PMC6463911.”

Response: In agreement with the statistical approach of Kain V. et al. (PMC6463911) we also used Kruskal-Wallis test to determine differences in the numbers of gut microorganisms. However, this work was not aimed at the integration of microbiome data with infarct size. We plan more extensive integration of data after metagenomic profiling of microbiota. With the data on bacterial count obtained with conventional RT-PCR it seems preliminary to perform exact statistical matching of microbiome data and cardiac outcomes.

9. Critique: “This multilayered study needs clarified presentation which lacking in the entire manuscript.”

Response: We agree with the Reviewer. We have included Figure 1 in the current revised version of the manuscript in order to clarify the experimental design (p. 9). 

10. Critique: “In discussion suggest, how obesity impacted the overall cardiac dysregulation of metabolism in-vivo tissue rather than ex-vivo.”

Response: Since we have not studied myocardial tolerance to ischemia-reperfusion injury in vivo in the present study, we can only conclude that diet-induced obesity in our study had no effect on myocardial tolerance to ischemia-reperfusion injury in our isolated heart model. We have included the comment on this issue in the Discussion section (p. 19). 

Response to the comments by Reviewer 2

First, we would like to thank the reviewer for the constructive comments about our manuscript. Although some points are correctly criticized, we believe that the findings of this study are relevant to the scope of the journal and will be of interest to its readership.

1. Critique: “Overall, the novelty of the current study is limited since cardioprotective effects of TTC in an experimental myocardial infarction model has been previously postulated.”

Response: We agree with the reviewer that cardioprotective effect of TTC has been previously postulated in healthy animals. The new finding provided by our study is the reversal of this cardioprotective effect in the animals with obesity and colitis.

2. Critique: “Please explain why despite differences in the extent of myocardial injury upon treatment with tetracycline, the hemodynamic parameters are not affected.”

Response: According to our experience, myocardial infarct size and functional performance of the isolated heart subjected to ischemia-reperfusion are not always showing parallel changes, especially when the extent of cardioprotection is not very strong. In this study, infarct size has been found to be more sensitive marker of cardioprotection than LV function. Although we have found some tendency towards lower values of LVEDP in TTC group, it was not statistically significant because of relatively high SD values.

3. Critique: “What are putative underlying mechanisms of TTC-mediated cardiac protection in lean mice and which mechanisms are dysregulated leading to detrimental effects of TTC in chemical induced colitis.» 

Response: The putative underlying mechanisms of TTC-mediated cardiac protection are summarized in the Discussion section (p. 17-18). The reversal of cardioprotective effect of TTC observed in the animals with obesity and colitis could be attributed to dramatically elevated levels of lipopolysaccharide, TNF-α and MCP-1 in these comorbid animals. The literature data suggest that these changes could neutralize the benefits of cardioprotection (discussed on p. 21, Discussion section). 

4. Critique: “Measuring SCFAs in the plasma is not expedient, since most of the SCFAs are metabolized in the liver before reaching the circulation. Thus, to assess changed in SCFA production, it is recommended to measure SCFAs in the feces. Protective effects of SCFAs are considered to be based on local paracrine mechanisms in the intestine (such as intestinal immune homeostasis) with secondary systemic consequences.» 

Response: In this study, we were interested to analyze the serum levels of short chain fatty acids (SCFAs) as potential direct preconditioning factors inducing myocardial tolerance to ischemia-reperfusion injury. However, this hypothesis has not been confirmed since we’ve failed to show any associations between SCFA levels in serum and infarct size limitation.

---

## [Decision Letter · Decision Letter 1]

29 Oct 2019

PONE-D-19-22988R1

Effects of tetracycline on myocardial infarct size in obese rats with chemically-induced colitis

PLOS ONE

Dear Dr. Galagudza,

Thank you for submitting your manuscript to PLOS ONE. After careful consideration, we feel that it has merit but does not fully meet PLOS ONE’s publication criteria as it currently stands. Therefore, we invite you to submit a revised version of the manuscript with the small changes in the text suggested by the two reviewers.

We would appreciate receiving your revised manuscript by Dec 13 2019 11:59PM. To enhance the reproducibility of your results, we recommend that if applicable you deposit your laboratory protocols in protocols.io, where a protocol can be assigned its own identifier (DOI) such that it can be cited independently in the future. For instructions see: http://journals.plos.org/plosone/s/submission-guidelines#loc-laboratory-protocols

We look forward to receiving your revised manuscript.

Kind regards,

Michael Bader

Academic Editor

PLOS ONE

Reviewers' comments:

Reviewer's Responses to Questions

**Comments to the Author**

1. If the authors have adequately addressed your comments raised in a previous round of review and you feel that this manuscript is now acceptable for publication, you may indicate that here to bypass the “Comments to the Author” section, enter your conflict of interest statement in the “Confidential to Editor” section, and submit your "Accept" recommendation.

Reviewer #1: All comments have been addressed

Reviewer #2: All comments have been addressed

2. Is the manuscript technically sound, and do the data support the conclusions?

Reviewer #1: Yes

Reviewer #2: Yes

3. Has the statistical analysis been performed appropriately and rigorously? 

Reviewer #1: Yes

Reviewer #2: Yes

4. Have the authors made all data underlying the findings in their manuscript fully available?

Reviewer #1: Yes

Reviewer #2: Yes

5. Is the manuscript presented in an intelligible fashion and written in standard English?

Reviewer #1: Yes

Reviewer #2: Yes

6. Review Comments to the Author

Reviewer #1: Majority comments answered, failed to update text in terms of recent changes of microbiome in context of obesity which is common comorbidity in patients, add this specific references suggested in original first critiques/comments.

Reviewer #2: The authors have satisfactorily discussed my previous points. Please address in the discussion section that the changes in myocardial infarct size were not reflected by changes in performance of the isolated heart subjected to ischemia-reperfusion.

7. PLOS authors have the option to publish the peer review history of their article (what does this mean?). If published, this will include your full peer review and any attached files.

Reviewer #1: No

Reviewer #2: No

---

## [Author Response · Author response to Decision Letter 1]

30 Oct 2019

Response to the comments by Reviewer 1

We would like to thank the reviewer for the constructive comments on the manuscript.

1. Critique: “Majority comments answered, failed to update text in terms of recent changes of microbiome in context of obesity which is common comorbidity in patients, add this specific references suggested in original first critiques/comments.” 

Response: We have modified the discussion (p. 21) and added the relevant reference according to the reviewer’s advice.

Response to the comments by Reviewer 2

We would like to thank the reviewer for the fair criticism of the manuscript.

1. Critique: “Please address in the discussion section that the changes in myocardial infarct size were not reflected by changes in performance of the isolated heart subjected to ischemia-reperfusion.”

Response: This issue is now discussed in the Discussion section (please see changes in p. 17).

---

## [Editor Report · Decision Letter 2]

31 Oct 2019

Effects of tetracycline on myocardial infarct size in obese rats with chemically-induced colitis

PONE-D-19-22988R2

Dear Dr. Galagudza,

We are pleased to inform you that your manuscript has been judged scientifically suitable for publication and will be formally accepted for publication once it complies with all outstanding technical requirements.

With kind regards,

Michael Bader

Academic Editor

PLOS ONE
---

## [Editor Report · Acceptance letter]

5 Nov 2019

PONE-D-19-22988R2 

Effects of tetracycline on myocardial infarct size in obese rats with chemically-induced colitis 

Dear Dr. Galagudza:

I am pleased to inform you that your manuscript has been deemed suitable for publication in PLOS ONE. Congratulations! Your manuscript is now with our production department. 

With kind regards,

on behalf of

Prof. Michael Bader 

Academic Editor

PLOS ONE